# Personalized recurrence risk assessment following the birth of a child with a pathogenic de novo mutation

Marie Bernkopf [1,2,3,24], Ummi B. Abdullah[1,2,24], Stephen J. Bush [1], Katherine A. Wood[1], Sahar Ghaffari[1], Eleni Giannoulatou[4], Nils Koelling [1], Geoffrey J. Maher[1], Loïc M. Thibaut [5], Jonathan Williams [6], Edward M. Blair[2,7], Fiona Blanco Kelly[7], Angela Bloss[7], Emma Burkitt-Wright[8,9], Natalie Canham[10], Alexander T. Deng [11], Abhijit Dixit[12], Jacqueline Eason[12], Frances Elmslie[13], Alice Gardham[14], Eleanor Hay[15], Muriel Holder[11], Tessa Homfray[13], Jane A. Hurst[15], Diana Johnson[16], Wendy D. Jones[15], Usha Kini[2,7], Emma Kivuva[17], Ajith Kumar [15], Melissa M. Lees[15], Harry G. Leitch[12,18], Jenny E. V. Morton[19], Andrea H. Németh[7,20], Shwetha Ramachandrappa [11], Katherine Saunders[7], Deborah J. Shears[7], Lucy Side[21], Miranda Splitt[22], Alison Stewart[16], Helen Stewart[7], Mohnish Suri[12], Penny Clouston[6], Robert W. Davies[23], Andrew O. M. Wilkie [1,2,25] & Anne Goriely [1,2,25] ✉

Following the diagnosis of a paediatric disorder caused by an apparently de novo mutation, a recurrence risk of 1–2% is frequently quoted due to the possibility of parental germline mosaicism; but for any specific couple, this figure is usually incorrect. We present a systematic approach to providing individualized recurrence risk. By combining locus-specific sequencing of multiple tissues to detect occult mosaicism with long-read sequencing to determine the parent-of-origin of the mutation, we show that we can stratify the majority of couples into one of seven discrete categories associated with substantially different risks to future offspring. Among 58 families with a single affected offspring (representing 59 de novo mutations in 49 genes), the recurrence risk for 35 (59%) was decreased below 0.1%, but increased owing to parental mixed mosaicism for 5 (9%)—that could be quantified in semen for paternal cases (recurrence risks of 5.6–12.1%). Implementation of this strategy offers the prospect of driving a major transformation in the practice of genetic counselling.

The birth of a child with a serious clinical disorder to a healthy couple with no previous family history is a life-changing event. Added to the challenges posed by caring for their child, is the anxiety that their future children could be similarly affected. Whilst robust frameworks for addressing this possibility are increasingly available for common chromosomal abnormalities and recessive monogenic diseases, no systematic approach has been developed for dominant disorders caused by apparently de novo mutations (DNMs). Such disorders are collectively common, estimated to affect at least 1 in 295 births[1], but extremely heterogeneous; for example, mutations in over 650 genes are currently recognized to cause developmental disorders through a dominant mechanism of action[1,2]. The need to address this issue has been made more pressing by the success over the past decade of next-generation sequencing (NGS) technologies in identifying DNMs, leading to a deluge of new causative genes and diagnoses.

**Fig. 1 | Stratification of DNMs into seven categories.** Establishing the origin (paternal [blue], maternal [pink] or post-zygotic [proband, green]), and timing of the mutational events (purple colour indicates mutant cells), yields widely different recurrence risks in different families. See main text, Supplementary Fig. S1 and Supplementary Note 1.

The implementation of NGS technologies across large populations has contributed to a better understanding of the patterns of occurrence of DNMs. It is now well established that DNMs are rare events (spontaneous human mutation rate is ~1.2 × 10$^{-8}$ per bp, per generation), mainly occurring as "one-off" copying errors during sperm production, or less frequently in oocytes[3,4]. While in these instances, the risk of recurrence for future siblings would be negligible, DNMs can also occur post-zygotically (either in one of the two clinically unaffected parents, or in the affected child) leading to a mosaic genotype that alters the recurrence risk. Mosaicism populating multiple germinal cells in the ovaries or testes (arising during one of the parent's own embryonic development), termed gonadal (or germline) mosaicism, may be associated with a substantial recurrence risk for further offspring, reaching up to 50% in some cases; by contrast, demonstration of post-zygotic mosaicism in the offspring would remove the risk of sibling recurrence[5].

Although mosaicism has long been recognized as a source of DNMs, few studies have attempted (or had the power) to define its exact contribution to spontaneous disease. Overall, current NGS methods used to identify DNMs rely on mother-father-proband trio sequencing and are poorly suited for detection of mosaic cases— either for cases of low-level (parental) mosaics[6], or to distinguish high-level variant allele frequency (VAF) from constitutional (50%) presentation in post-zygotic (proband) cases[5,7]. For example, the limit of VAF sensitivity of whole exome (WES) or whole genome (WGS) sequencing, which are typically performed at a depth of 25–30×, is ~10–15%, similar to that of dideoxy-sequencing[6,8]. Moreover, routine genetic analysis relies on the interrogation of a single somatic tissue (blood or saliva), which is not adequate to identify mosaicism in parental gametes or variable VAF in a proband's tissues.

The recognition that the tissue distribution and VAF of a DNM are determined by the timing at which it first occurred, allows us to identify three key time points during development with different predicted presentations: (1) very early in development—before the segregation of germline and somatic lineages at ~day 14 of human embryogenesis, yielding cases of mixed (somatic and gonadal) mosaicism; (2) post-15 days of development in the germline lineage, resulting in confined gonadal mosaicism; (3) or much later in the developing or adult gonad, yielding a "one-off" mutation (Supplementary Fig. S1). Furthermore, by taking into account the individual in whom the DNM originated (mother, father, or affected child), it becomes possible to distinguish a total of seven scenarios whereby a DNM can occur (Fig. 1). The overall relative prevalence of these seven scenarios can be estimated quite accurately based on previous analyses of the parental origin of DNMs and the prevalence of mosaicism from population studies (see Supplementary Note 1).

Here, we describe the approach and results of the PREGCARE (PREcision Genetic Counselling And REproduction) research study, which developed a systematic strategy to categorize pathogenic DNMs in a mixed clinical population of 60 couples who had one or more children diagnosed with a serious developmental disorder caused by an apparent DNM, and were seeking individualized reproductive counselling about recurrence risk in a future pregnancy. By combining targeted ultra-deep Next Generation Sequencing (NGS) of multiple tissues to detect occult mosaicism with locus-specific haplotyping to determine the parent-of-origin of the DNM, we show that we can reliably stratify individual couples into discrete categories that are

**1. Verification of the family relationship (MIP assay)**

**2. Targeted ultra deep-NGS of each biological sample of the trio + 3 controls (performed in triplicate)**

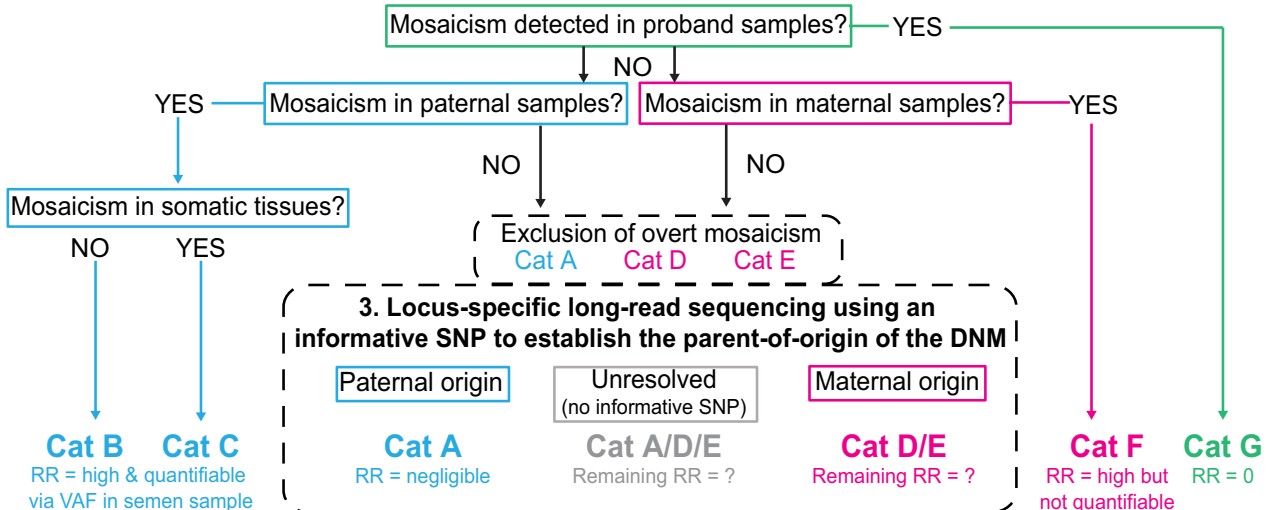

**Fig. 2 | Flow chart describing the three-tier sample analysis in the PREGCARE study.** Following collection of up to 14 different biological samples per family and verification of the familial relationships between the 3 individuals of the trio, the DNM site was deep-sequenced in all family samples (performed in triplicate reactions together with 3 unrelated controls) to detect low levels of parental mosaicism or instances of post-zygotic mosaicism in the proband. For those families without evidence of overt mosaicism, haplotyping using long-read sequencing (MinION platform from Oxford Nanopore Technologies (ONT)) was performed to resolve the parental origin of the DNM and further stratify the recurrence risk (RR). Refer to Fig. 1 for category (Cat) description.

associated with substantially different risks to the offspring. This personalized approach to recurrence risk assessment offered *prior* to a new pregnancy should provide reassurance to the majority of couples in whom the risk is very low or negligible, and help to focus resources on the minority of families at increased recurrence risk.

## Results

### Population sampled

Following ethical approval we recruited, through the network of Clinical Genetics centres in England, 60 couples who had one or more children (or fetuses) affected by a serious disorder caused by an identified DNM, which was not present in the parents' DNA on routine clinical analysis (see Methods; Supplementary Data 1). Two families (FAM17 and FAM60) had three affected siblings/pregnancies, indicating that one of the parents must be a gonadal mosaic, but routine diagnostic analysis performed on parental blood DNA had failed to identify the parent-of-origin. To eliminate ascertainment bias, these two families are excluded from the quantitative presentation of the data but included in the specific analysis of mosaicism. Hence, our primary cohort comprises data from 58 parent-child trios, including one trio with two different pathogenic DNMs (FAM12). These 59 DNMs comprised 40 single nucleotide substitutions, 14 small (1–2 nucleotides) indels, and 5 larger (4–44 nucleotides) indels in 49 different genes, providing a broad and representative spectrum of pathogenic molecular lesions encountered in clinical practice (Supplementary Data 1).

### Deep-sequencing of multiple tissues identifies mosaic cases

Four of the seven categories shown in Fig. 1 (i.e., categories B, C, F, and G) involve mosaic states that can be directly identified and distinguished by Deep-NGS of tissues collected from the family trio. Therefore we obtained up to 14 biological family samples (child: blood, buccal mucosa left + right; mother and father: blood, saliva, buccal mucosa left + right, urine; plus paternal semen) to seek evidence of mosaicism. Collection of parent samples was designed to include all three embryonic germ layers (ectoderm, buccal; mesoderm, blood; endoderm, urine), plus germline in the father (Supplementary Fig. S1).

The overall strategy deployed for the analysis is shown in Fig. 2. Following verification of sample relationships and parentage in each family using a panel of bespoke molecular inversion probes (MIPs) targeting 168 common single nucleotide polymorphisms (SNPs), we designed a custom PCR assay covering 49–266 bp around the family-specific DNM site and performed triplicate reactions from each available tissue, and three unrelated control DNAs, before undertaking Deep-NGS (target depth ≥ 5,000x in order to detect VAFs <1%) on the Illumina MiSeq platform. Reads were processed using amplimap[9] and VAF quantified at the genomic position of the DNM (see Methods). NGS was poorly suited to analyze two DNMs associated with the larger indels (a 44 bp deletion in FAM12b and a 35 bp duplication in FAM54). Hence, to rule out the possibility of occult mosaicism in these samples, we performed mutant allele-specific PCR on all available samples from the two trios (Supplementary Note 2).

Overall, Deep-NGS (and/or allele-specific PCR) identified 7/59 (11.9%) cases with strong evidence of mosaicism in one family member (Fig. 3; Supplementary Fig. S2 and Supplementary Data 2). These comprised DNMs belonging to Categories B (paternal gonadal mosaicism; FAM27), C (paternal mixed mosaicism; FAM34, FAM49, FAM58), F (maternal mixed mosaicism; FAM01, FAM50) and G (post-zygotic mosaicism in proband; FAM33). Analysis of the two additional families in which recurrence in siblings was already documented (FAM17, FAM60) showed that both were attributable to maternal mixed mosaicism (Fig. 3).

Identifying these mosaic families is particularly important, because whereas the recurrence risk associated with post-zygotic mosaicism (Category G) is effectively zero, the other three mosaic categories (B, C, F) are potentially associated with increased recurrence risks. While the offspring risk is not directly quantifiable for the maternal mosaics because of the inaccessibility of ovarian tissue, it could be quantified in the paternal mosaic cases via the VAF measured in sperm and ranged from 0.23% (FAM27) to 12.1% (FAM58) (F6 bars in Fig. 3; Supplementary Fig. S2).

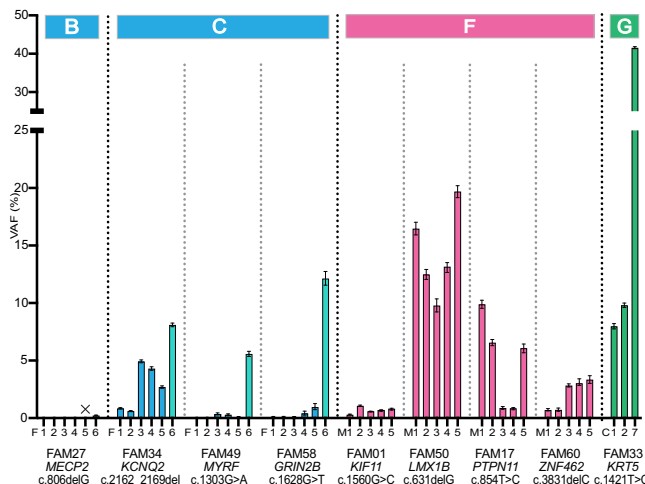

**Fig. 3 | Mutation levels observed in the families presenting with mosaicism.**
Variant allele frequencies (VAF) in different samples from the family member in whom mosaicism was detected by ultra-deep Illumina Sequencing (Deep-NGS). Family number, gene, cDNA coordinates of the DNM and the origin of the different samples are indicated on the x-axis in the same order for each family. The category classification is indicated at the top of the figure and the colours reflect the parental/post-zygotic origin of the DNM (blue [somatic paternal tissues], turquoise [paternal sperm], pink [somatic maternal tissues] or green [post-zygotic/proband tissues]). The × represents a sample failure. Full data for the other family members and controls are presented in Supplementary Fig. S2 and Supplementary Data 2. FAM17 and FAM60 are the two families with multiple affected pregnancies and belong to Category F (maternal mixed mosaicism); note the low VAF in the maternal blood samples (M3) for both families. Each bar represents a single VAF calculated from the sum of 3 technical replicates, then corrected using measurements from 3 unrelated controls (corrected VAFs, see "Methods"). Error bars represent the 95% binomial confidence intervals. Abbreviations: F father; M mother; C child; 1 = buccal mucosa (left); 2 = buccal mucosa (right); 3 = blood; 4 = saliva; 5 = urine; 6 = sperm; 7 = genomic DNA from original testing.

In seven of the eight parental mosaic cases, the level of mutation in the DNA derived from blood (the most widely used source of DNA for genetic analysis) obtained from the transmitting parent was below 5% (F3 and M3 values in Fig. 3; Supplementary Data 2). Such VAFs would be impossible to detect systematically using standard diagnostic NGS read depths (~25–30×) or dideoxy-sequencing, illustrating the importance of deep-sequencing and the value of collecting additional tissue samples to increase sensitivity for ascertaining occult mosaicism (Supplementary Note 3; Supplementary Fig. S3). Importantly, in the three paternal cases of mixed mosaicism the level of the cognate DNM in sperm (5.6–12.1%) was substantially higher than in any of the other tissues sampled and variability in mutation levels was present between different somatic tissues, with no one tissue providing a reliable indicator of the level in sperm (Fig. 3). In the single identified instance of paternal confined gonadal mosaicism, a relatively low level of sperm mutation was observed (0.23%), consistent with a slightly later timing of mutational origin (Supplementary Fig. S1) and in line with empiric data on mutation levels in sperm[10–14].

Also of note, the VAFs for the proband samples from FAM33 with the post-zygotic mutation (C1, C2, and C7 values in Fig. 3) were markedly different across the tissues analyzed (blood 41.6%; buccal mucosa 8.0 and 9.8%), demonstrating the benefit of analyzing several tissue samples from an individual to distinguish post-zygotic mosaicism associated with high VAF levels from constitutional (50%) presentation.

## Long-read sequencing enables haplotype phasing and determination of the DNM parental origin

For the remaining 52 DNMs that did not classify into one of the four mosaic categories described above, further stratification was

attempted through haplotyping to determine the parental origin of the mutation (Fig. 2). For these families, only one category (Category E, maternal gonadal mosaicism) is associated with a recurrence risk to offspring (Fig. 1). Although it is not possible to distinguish Categories D and E (because oocytes are not accessible), most of the remaining DNMs (~80%) are predicted to belong to Category A (paternal one-off), which is associated with a negligible risk to offspring.

Parental origin could be inferred for two families without performing haplotyping: FAM26 (mutation in the X-linked *MID1* gene in a male proband, implying a maternal origin) and FAM54 (a 35 bp duplication in *MAGEL2*, a gene known to be maternally imprinted and for which pathogenic mutations are exclusively paternal in origin[15]). To perform haplotyping of the other 50 DNMs, we sought an informative SNP or other variant in close proximity to the DNM, to enable phasing of the parental alleles. In the most common informative scenario, the child is heterozygous for the SNP (genotype AB) whereas one of the parents is homozygous (genotype AA or BB), making it possible for the inherited parental chromosomes to be distinguished. In three cases, an informative SNP was present in the amplicon used for the targeted Deep-NGS assay, enabling the parental origin to be determined directly by examining the phase of the DNM from the Illumina reads. To haplotype the 47 remaining DNMs, we designed two long PCR products extending away on either side of the DNM (total genomic region covered ~7–30 kb) and sequenced the resulting fragments for the three family members using the MinION platform from Oxford Nanopore Technology (ONT). Reads for each trio were processed and analyzed with an in-house custom pipeline combining Medaka and pile-up processing (see "Methods" and Supplementary Note 4). This haplotyping strategy was successful in the majority (38/47) of cases (Supplementary Data 3), including three families (FAM11, FAM38, FAM67) that required a more complex analysis involving two SNPs to distinguish the parental alleles. In one of these (FAM38), due to the local genomic context of the DNM (a single G-nucleotide deletion within a homopolymeric region), phasing by direct analysis of long-read sequencing traces could not be resolved. Nevertheless, this approach identified an informative SNP in the proband which was used to design a bespoke allele-specific PCR and determine the DNM parental origin (Supplementary Note 5 and Supplementary Data 3B).

Overall, parental origin could be established for 82.7% of DNM (43/52), which included 34 DNMs of paternal origin (79%) and 9 (21%) present on the maternally-derived allele (Fig. 4)—a result in line with the expected ~4:1 male to female ratio of mutational origin[3] (Supplementary Note 1).

## Combining Deep-NGS and haplotyping by long-read sequencing allows category stratification and individualized recurrence risk estimation

Having singled out the mosaic cases by Deep-NGS of multiple familial tissues, the particular value of the combined approach of deep-sequencing of semen samples with haplotyping is to identify those families in which the DNM is paternal in origin (34/52), as they belong to Category A (Figs. 1 and 2). Deep-NGS of sperm of these paternal mutations allows measurement of the VAF and derivation of the upper confidence limit for the level of DNM present in sperm (Supplementary Data 2). As NGS is subject to background sequencing errors, we corrected the raw VAF values using measurements from the three unrelated control samples. The corrected VAFs were estimated by numerically maximizing their marginal likelihood, and 95% confidence intervals were obtained by using profile likelihood (see "Methods" for details and Supplementary Note 3). For Category A samples, the upper bound (95% CI) of the VAF measured by Deep-NGS in sperm was below 0.05% in all cases (Supplementary Data 2; Fig. 4). These data point to DNMs in this category having originated as 'one-off' events during late gonadal development or adult spermatogenesis.

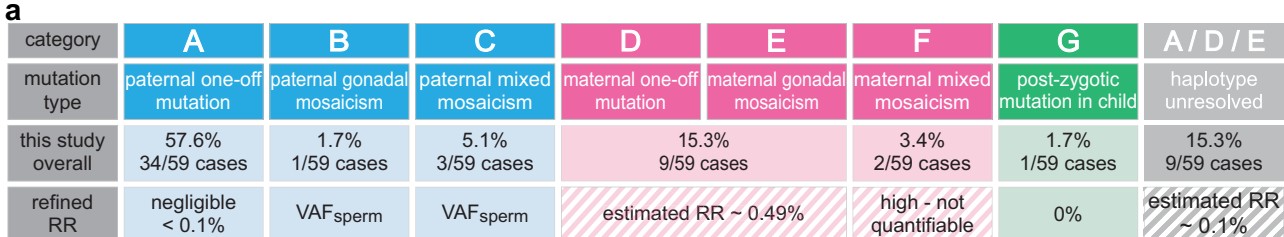

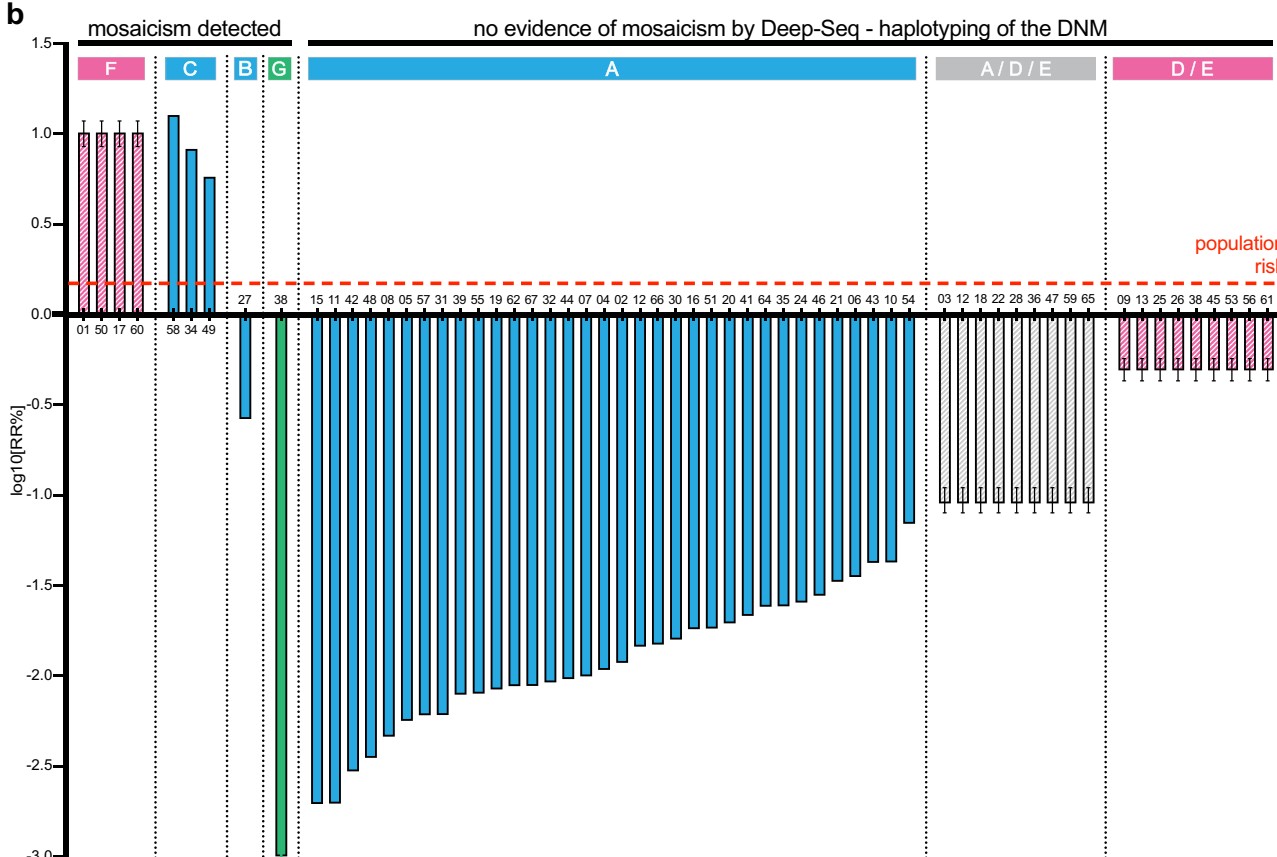

**Fig. 4 | Overview of the results of the PREGCARE study showing refinement of individual recurrence risk for all families. a** Summary table of the PREGCARE results for the 59 DNMs analyzed in this study and overview of the refined recurrence risk (RR). **b** Personalized recurrence risk (RR%) estimates for each of the 60 families (61 DNMs) enrolled in the PREGCARE study represented on a logarithmic scale. The red dotted line represents the generic population RR given to couples who have had a child with a DNM (-1.5%). Individual family numbers are indicated on the *x*-axis. Each bar represents the new refined RR for a given family enrolled into the PREGCARE study. For ease of visualization, bars are coloured according to the origin of the DNM [blue (paternal), pink (maternal), green (post-zygotic/proband) or grey (the parental origin could not be resolved)]. The RR can be quantified for

DNMs of paternal (Categories A–C, via semen analysis, blue) and post-zygotic (Category G, green) origin (represented by block colours); note that to be conservative in our estimates of RR, the plotted bars represent the upper 95% binomial CI from the corrected VAF measured in sperm samples by Deep-NGS for each of the paternally-derived DNMs (Supplementary Data 2). The RR can only be estimated for maternal (pink) or haplotype-unresolved cases (grey). These estimates are represented by stripes, with error bars representing the upper and lower 95% CI—see Supplementary Note 6 for details on estimate calculations. Note that the DNM of FAM54 was analyzed by allele-specific PCR (Supplementary Note 2) and that Category F includes the two additional families with multiple affected pregnancies (FAM17 and FAM60).

**Estimation of the remaining risk for maternal cases (Categories D/E) and cases of unresolved parental origin (Categories A/D/E)**

In 9/52 (17.3%) families, the haplotyping revealed a maternal origin of the DNM. In these cases, the negative findings from Deep-NGS of maternal somatic tissues (i.e., DNM of maternal origin but with no evidence of somatic mosaicism) do not allow Categories D and E to be distinguished (Figs. 1–2 and 4). Nevertheless, as only Category E poses a risk of recurrence in a future pregnancy, the recurrence risk of the combined maternal D/E categories can be estimated. The relative prevalence of a DNM belonging to Category E (maternal gonadal mosaicism; 2% of cases as shown in Fig. 1) rather than to Category D ('one-off' maternal event; 15% of DNMs) implies that on

average only 1 in every 8 or 9 maternal DNMs will have originated in the early developing germline lineage. While mosaic VAFs have not been directly quantified in ovaries owing to their experimental inaccessibility, these early events are predicted to be similar in magnitude for ovaries and testes because germline lineages are specified several weeks prior to sex determination (Supplementary Note 1)[4]. Hence, we used the VAFs observed for paternal confined mosaics obtained from sperm WGS data[11] as a proxy to estimate the average VAFs associated with Category E. Combining the proportion of cases (2:15) and associated VAFs for confined gonadal mosaicism, we obtained a recurrence risk estimate for the combined maternal Categories D/E of 0.49% (95%CI: 0.43–0.57%) (Supplementary

Note 6), a modest reduction compared to the population average (Figs. 1 and 4).

Finally, in a further 9 families, despite sequencing a ~10–27 (mean 20.6) kb region around the DNM in the proband, no informative SNP could be identified in the long-read sequencing data. As the parental origin of the DNM could not be assigned, the mutation could belong to any of Categories A, D, or E (Fig. 1). Because the majority of zygotic DNMs are predicted to be sporadic (see Fig. 1: Categories A (71% of cases) or D (15%)), the remaining risk (associated with Category E (2%)) for these couples can be estimated by combining the relative contribution of Category E cases (2:86) and the average VAF observed for confined gonadal mosaicism[11]. For a DNM with an unresolved parent of origin (for which parental mixed mosaicism has been excluded by Deep-NGS), the recurrence risk is estimated to be 0.09% (95%CI: 0.08–0.11%), a reduction of approximately 10-fold compared to the population risk baseline (Fig. 4; Supplementary Note 6).

## Discussion

We have applied a general framework to analyze systematically and at scale, the origins of pathogenic DNMs presenting in a clinical setting. This work addresses a stark unmet clinical need to improve genetic counselling for couples who have had a child affected by a disorder caused by a DNM−a situation faced by almost a million parents annually−in order to provide them with a personalized risk assessment prior to a new pregnancy. The current standard of care, which is to provide these couples with a recurrence risk of ~1–2%, is unsatisfactory, both because this figure is nearly always wrong (as illustrated by Fig. 4), but also because of the uncertainty it raises for the couple in the complex decision process of whether to extend their family. It is well documented that couples' attitudes to reproductive risk vary widely[16]: some will view the 1–2% risk as small and others would not contemplate extending their family in the face of any risk. In addition, while in many healthcare settings there may be the option of a prenatal diagnostic procedure (chorionic villus biopsy or amniocentesis), this is associated with a small risk of miscarriage (currently estimated as ~0.2–0.5% for each procedure[17–19]) and may not be ethically acceptable to some couples. Owing to a combination of cost and technical challenges, prenatal procedures that are non-invasive (assay of free fetal DNA from maternal blood sample) or those which avoid the possibility of termination of pregnancy in the event of recurrence (preimplantation genetic testing for monogenic disorders [PGT-M]), are not available in most public healthcare settings. For example, in the UK the eligibility threshold for PGT-M is a risk >10% of having a child with a serious genetic condition, which excludes the parents of children with DNMs even though some couples will have a risk higher than this.

Over recent years several pioneering studies on DNM origins have provided a solid framework to quantify the relative contribution of different mutational processes to DNMs (Fig. 1; Supplementary Notes 1 and 6)[3–5,20–22]. We designed the PREGCARE study based on this framework, with the dual aims to seek evidence for mosaicism in each member of the parent-child trio (Deep-NGS), and to stratify the risk based on the likely timing and parental origin of the DNM. Important aspects of the study design include the recognition that (1) clinically-relevant mosaicism is caused by early embryonic mutations, that present either in both soma and germline (mixed mosaicism) or the germline only (confined gonadal mosaicism) and affect males and females equally−because they originate before sex determination; (2) sampling of multiple tissues of different embryonic origins increases the likelihood of detecting instances of mixed mosaicism in parents (or post-zygotic events in the proband); and (3) analysis of a paternal semen sample allows direct quantification of risk for paternally-derived DNMs, which are anticipated to represent ~¾ of cases. Although the female germline is not accessible to direct analysis, data about the prevalence and VAF anticipated for maternal mosaic cases can be inferred from sperm data[11]. Moreover, the relative risks of mixed vs.

confined gonadal mosaic events can be estimated based on data from deep-WGS of paired blood and sperm samples[10,11], which have shown that the average VAF measured in sperm for mixed mosaic variants is ~8%, while that of 'sperm-only' DNMs is ~3%[11] (Supplementary Note 6). These estimates reflect the fact that because mixed mosaicism is caused by very early mutational events, the variants are present at higher VAFs and exhibit a wider tissue distribution (Supplementary Fig. S1). Moreover, the rate of spontaneous mutations may be elevated during the first embryonic divisions[23,24]. Hence, mixed mosaic cases likely contribute to most of the recurrence risk in the next generation, and identifying them by Deep-NGS of somatic tissues (and semen) represents an efficient way to single out the couples at higher risk.

Our systematic analysis of a clinical series of 59 DNMs shows a very good correspondence between the distribution of DNMs across the seven different categories for the families analyzed, compared to that anticipated from population estimates (Supplementary Note 1; Figs. 1 and 4A). In our cohort, which consists of clinically-ascertained cases, DNMs originated from occult parental mosaicism in ~10% (6/59) of cases (95% confidence limits 3.8–20.8%). For five families it was detectable in the transmitting parent's somatic tissues−although present at low VAFs in blood−illustrating the value of collecting additional tissue samples and the importance of performing Deep-NGS (~5000×) using technical replicates to increase sensitivity for ascertaining occult mosaicism (Supplementary Note 3).

Figure 4, which summarizes our overall findings, shows that we achieved risk alteration for individual couples over more than three $\log_{10}$ orders of magnitude: for 54/59 DNMs, the risk was reduced compared to the population baseline risk, and for 5/59 (the mixed mosaics), it was likely increased (but only quantifiable in the 3/5 paternal cases). Encouraging though these data are, we acknowledge several barriers before considering clinical translation of this work. The first hurdle relates to technical implementation of individualized recurrence risk measurement in a clinical setting, which requires robust laboratory methods and will be challenging as a DNM-specific custom assay will be required for most families. In this study we used two methods, targeted Deep-NGS and locus-specific long-read haplotyping, which provide complementary information. Deep-NGS is highly effective in singling out couples at high recurrence risk, whereas haplotyping is essential to generate most of the very low recurrence risks and reassure the majority of couples that they belong to Category A. The Illumina platform used for Deep-NGS is technically straightforward and the associated calling pipelines are readily available in most diagnostic settings. Of note, for efficient evaluation of reproductive risk, the source of tissue samples is a major consideration. While semen provides the ideal tissue for determining reproductive risks directly, at present the prevailing working practices of clinical genetics do not include routine semen collection. Our view is that this work and that of others[10–13,25] provides clear evidence to promote much more widespread analysis of this material (as is standard, for example, in fertility clinics). Overall, clinical implementation of targeted Deep-NGS of a few key tissues (see Supplementary Note 7) from the trio (including blood, buccal brushings, and paternal semen) should be readily achieved and would identify most of the high-risk cases, therefore reducing the risk of mosaicism presentation for the remaining couples (Categories A/D/E) to ~0.1% (Supplementary Note 6).

To further refine the remaining risk in non-mosaic families, we performed locus-specific long-read sequencing on the ONT platform as a second method in this study. Although harder to scale and process than short-read Illumina NGS data, ONT showed good potential for implementation in diagnostics, but is not currently approved for use in most clinical settings. Independently of technical considerations, one major limitation of this approach, which led to a substantial minority (15%) of unresolved cases in this study, is the requirement for the presence of a heterozygous SNP in the vicinity of the DNM to distinguish the two parental alleles in the proband (Supplementary Note 4).

Implementation of novel ultra-long read and/or methylation-aware WGS methods will facilitate systematic parent-of-origin assignment of DNMs[26,27].

Another potential barrier to clinical implementation relates to how these refined risks are viewed by couples and whether changes in risk actually result in altered decision-making. Concerning the accuracy of our risk estimations, among the 61 DNMs analyzed, in only 39 do we consider the risks to be reasonably accurate; these include the 38 DNMs shown to be paternally originating (Categories A–C), in which we could directly measure levels of mutation in sperm, and the single post-zygotic case (Category G) (Fig. 4). In 36/39 we reported a risk lower than baseline, while in the three mixed mosaic cases it was increased (to 5.6, 8.1, and 12.1%). By contrast, in the remaining cases shown either to be of maternal origin (Categories D–F; 13/61) or haplotype-unresolved (Categories A/D/E; 9/61), the risk estimation has been refined but remains inaccurate and may be viewed differently by parents and healthcare professionals. Even in proven cases of maternal mixed mosaicism, VAFs in somatic tissues are poor predictors for the germline, as illustrated by the two families with multiple recurrences in whom we detected relatively low VAF in maternal somatic tissues (maximum of 3.3 and 9.9% in the samples analyzed), despite three affected pregnancies in each sibship (Fig. 3). Nevertheless, detection of mixed mosaicism in maternal tissues will warrant caution in future pregnancy and it should also be noted that some diagnostic options may be more complicated for these families because of the unsuitability of non-invasive prenatal testing via analysis of cell-free fetal DNA in maternal plasma[28].

In those cases where somatic mosaicism has been excluded but the DNM is proven or possibly maternal in origin, the risk of maternal gonadal mosaicism (Category E, Fig. 1) may remain an important factor in decision-making, despite the relative reduction in risk for these subcategories (Category E represents ~2:15 maternally-proven DNM and ~2:86 haplotype-unresolved DNM with an estimated average VAF of ~3%).

An interesting illustration from this work of the complexity of recurrence risk counselling is provided by the case of paternal confined gonadal mosaicism (Category B) in FAM27, in which the risk for the *MECP2* mutation was found to be 0.23% (95% CI: 0.19–0.26%), over 4-fold lower than the 1–2% baseline population risk. How to counsel a couple in this situation, where stratification to the "at risk" Category B predicts increased caution, remains difficult. For example, the level of risk for FAM27 is only modestly lower than the current UK recommendation for 'higher risk' with respect to the threshold value for carrying a fetus with Down syndrome following routine screening (0.66%), at which non-invasive prenatal testing is currently recommended[29].

In conclusion, we show that providing pre-conception recurrence risk assessment to couples who have had a child with a DNM can be achieved, and offers the prospect of driving a major transformation in the practice of genetic counselling. Our data demonstrate that for all couples, it is possible to refine the risk of having another affected child with the same DNM and in the majority of cases the risk is in fact negligible, potentially reducing anxiety and the need for expensive pre-implantation or prenatal diagnostic options. For couples in whom we detected overt mosaicism, the risk is increased (and quantifiable through sperm analysis for the paternal cases). Providing evidence-based estimation of the actual risk will allow these couples to be prioritized for further investigations and support, enabling them to make informed choices about the different diagnostic options available to them.

## Methods

### Recruitment into the PREGCARE study
The PREGCARE (PREcision Genetic Counselling And REproduction) study was approved by the London−Queen Square Research Ethics Committee under the reference number 17/LO/1025 (IRAS reference: 225264). Couples with one (or multiple) children, stillbirths or terminated pregnancies affected by a likely pathogenic de novo mutation (DNM) and who were potentially interested in personalized transmission risk assessment for future pregnancies, were invited to participate by healthcare professionals during routine clinical genetic consultation. A DNM was defined as a single-nucleotide or small insertion-deletion variant detected in the proband that was absent in the parents' DNA on routine diagnostic genetic analysis. Because the increased prevalence of most DNMs in so-called paternal age-effect genes (including *FGFR2, FGFR3, HRAS, KRAS, PTPN11, RET*) is known to be associated with age-related expansion of mutant clones in testes (i.e., selfish spermatogonial selection), these carry a reduced risk of mosaic presentation[30]. Hence DNMs in these six genes were excluded from this research study, unless there were multiple affected pregnancies. Couples where the mother was pregnant at the time of sample collection, those who were not both the biological parents of the affected child, or either the biological mother or father did not consent to participate, were also excluded.

Recruitment and sample collection took place at 13 of the 17 participating National Health Service (NHS) Trusts in England, UK.

### Sample collection
Families interested in participating in the study were sent a box containing kits and instructions for collection at home of 2 ml saliva (Oragene DNA, OG-500, DNA-Genotek, Canada) and 50 ml morning midstream urine (Urine Collection And Preservation Tube, Norgen Biotek Corp., Canada) from both the mother and father, and an ejaculate of semen (following abstinence for three days before collection and stored at −20 °C) from the father. During the clinical visit, informed written consents were obtained from participants and further samples were collected from the three family members, including 5 ml peripheral blood (EDTA) from father and mother and buccal cells from the left and right inner cheek lining from mother, father, and the affected child using swabs (sterile PurFlock Ultra tip swab in dry transport tube, Puritan Medical Products, ME, USA). Samples and completed consent forms were sent at room temperature to the MRC Weatherall Institute of Molecular Medicine (Oxford) where they were witnessed-transferred and processed for extraction or long-term storage within 48 h of collection. Overall, a total of 67 boxes were dispatched and 60 families completed collection and consents and were enrolled into the study.

In addition, the child's genomic DNA originally used for the molecular diagnosis was requested from the NHS genetic laboratory. This sample had usually been extracted from the proband blood or, occasionally, fetal tissues, amniocentesis or chorionic villus sample (CVS) (for details see Supplementary Data 1).

### Sample processing and DNA extraction
Upon delivery of the box to the lab, the family samples were given a unique identifier and processed. The saliva samples were incubated at 50 °C for 60 min and then aliquoted. Blood samples were aliquoted as whole blood and isolated buffy coat. Urine samples were centrifuged at $2000 \times g$ for 10 min and the cell pellet rinsed with $1 \times$ phosphate-buffered saline (PBS) before storage. Semen samples were split into 50−100 μl volume aliquots which were rinsed with $1 \times$ PBS ($5000 \times g$ for 5 min). Mouth swabs were kept frozen until extraction, when they were resuspended into 100 μl PBS.

Genomic DNA was extracted from all the collected family samples (2 saliva lysates, 2 whole blood samples, 2 urine cell pellets, 6 buccal swabs, 1 semen lysate) on the Maxwell RSC Instrument using the Maxwell RSC Blood DNA kit (both Promega, WI, USA) and following

manufacturer's protocols. Aliquots of semen were pre-incubated in sperm lysis buffer (20 mM Tris HCl pH 8.0, 20 mM EDTA, 200 mM NaCl, 1% SDS) in the presence of proteinase K (250 μg/ml), dithiothreitol (DTT; 100 mM) and 0.6% SDS at 42 °C for 4–12 h. Concentrations of the final DNA eluates were assessed with standard fluorometric methods.

### Genotyping assay for verification of familial relationship using molecular inversion probes (smMIP assay)

To confirm the familial relationships of each trio, we used an in-house custom single-molecule molecular inversion probes (smMIPs) genotyping assay to capture common single nucleotide polymorphisms (SNPs) across all chromosomes (total of 290 smMIP probes targeting 154 autosomal, 14 X-linked and 57 Y-linked markers; for SNP details and probe sequences, see Supplementary Data 4A, B), following established smMIPs protocols[31]; this was followed by sample barcoding, library preparation and 2 × 151 bp paired-end sequencing on a MiSeq instrument (Illumina, CA, USA). For each family, DNA from the proband sample obtained from the original diagnostic laboratory (or if unavailable, buccal mucosa DNA), the maternal blood sample and the paternal semen sample were analyzed. Sequencing data were processed using the 'pileups snps' tool in the amplimap v0.4.9[9] pipeline with default settings (alignment to GRCh38.p12 with BWA, variant calling with GATK) to generate counts for the reference (REF) and alternate (ALT) alleles at each locus. Subsequently, the autosomal and X-linked SNP genotype for each individual of the family trio was recorded as Homozygous REF (AA), Heterozygous (AB) or Homozygous ALT (BB). For genotyping, SNPs were considered informative when the parents were homozygous (AA or BB) and the proband exhibited the expected genotype such as when Parent1/Parent2/Proband were AA/AA/AA, BB/BB/BB, AA/BB/AB. Other SNPs were analyzed to ensure there was no genotype discordance across the 3 family members.

### Ultra-deep Illumina sequencing (Deep-NGS) of DNM sites

Ultra-deep Illumina sequencing was performed in order to detect low levels of mosaicism in parental samples or post-zygotic mosaicism in the child. For each family-specific DNM, a pair of PCR primers tailed with generic CS1 (5′- ACACTGACGA-CATGGTTCTACA) and CS2 (5′- TACGGTAGCAGAGACTTGGTCT) sequence tags were designed to amplify a short genomic region (49–266 bp) around the DNM site; primer genomic locations (build GRCh38.p12), are provided in Supplementary Data 1. Each primer set was tested on control DNA with either High Fidelity Phusion or Q5 Polymerase (New England Biolabs, MA, USA) and PCR amplification was performed following the manufacturer's recommendations using 30 ng of genomic DNA from triplicates of up to 14 biological samples and three unrelated control DNAs in 10 μl PCR reactions, applying an initial denaturation step for 30 s at 98 °C, followed by 30 cycles of 10 s at 98 °C, 30 s at 68 °C, and 30 s at 72 °C, and 8 min at 72 °C as final extension step. Successful amplification was confirmed by running samples on an agarose gel. To construct the family-specific libraries, PCR-amplified fragments were diluted (1:100), further PCR amplified for 10 cycles using High Fidelity Phusion polymerase and 0.4 μM of unique 10 bp barcoded primers (PE1_CS1: 5′-AATGATACGGCGACCACCGAGATCT-CS1) −3′, PE2-BC-CS2: 5′- CAAGCAGAAGACGGCATACGAGAT[N$_{10}$]-CS2 -3′), where N$_{10}$ represents the generic Illumina Barcode sequences. The barcoded PCR products were visualized on a 2% agarose gel and mixed in near-equimolar ratio before being purified by gel extraction, and ultra-deep sequenced on a MiSeq (Illumina) instrument with 2 × 151 bp paired-end reads at an average depth of ~19,000× for each sample. For further details on library construction, please refer to Bernkopf et al.[25].

### Deep-NGS data analysis and determination of the observed variant allele frequency (VAF) at the DNM location

Illumina data were analyzed using amplimap[9], as above, to obtain both the variant allele frequency (VAF) of each family-specific mutation and the total count of >Q30 bases at the corresponding genomic position (GRCh38.p12) in each PCR replicate and sample. For each family-specific dataset, DNM VAFs observed in each sample were corrected, to account for the background alternate read counts observed in the control samples (false-positives) at the DNM genomic location. Let $k1$ and $k2$ be the number of alternate reads observed in the control and case, and $n1$ and $n2$ be the total number of reads observed in the control and case, respectively. Let $p$ denote the unobserved proportion of cells carrying a variant and let $q$ be the false-positive rate of the sequencing and variant-calling procedure.

The joint likelihood of $p$ and $q$ is defined as follows

$$\mathscr{L}(p,q)k1,n1,k2,n2) = B(k1; n1,q) \cdot B(k2; n2, p+(1-p) \cdot q) \quad (1)$$

where $B$ denotes the binomial probability mass function and $B(k; n, p)$ is the probability of observing $k$ successes in $n$ trials with success probability $p$. The first term corresponds to the probability mass of observing $k1$ false-positives in the control, and the second term corresponds to the probability mass of observing $k2$ alternate reads in the $p+(1-p) \cdot q$ case. The rate in the second term corresponds to the fact that a read identified as carrying the variant in the case is either a true positive (i.e., actually carrying the variant) with probability $p$ or a false positive (i.e., background noise, not carrying the variant but mistakenly identified as doing so) with probability $(1-p) \cdot q$.

We treated $q$ as a nuisance parameter and obtained the marginal likelihood of $p$ by numerically integrating the joint likelihood over $q$ using adaptive quadrature[32]. Finally, we obtained the maximum likelihood estimate of $p$ by numerically maximizing the marginal likelihood and obtained 95% confidence intervals using profile likelihood[33]. Scripts describing this analysis are available at github.com/sjbush/pregcare[34].

### Allele-specific PCRs

For two DNMs—a 44 bp deletion in *MECP2* in FAM12b and a 35 bp duplication in *MAGEL2* in FAM54—the regions were successfully amplified as described above, but the deep-sequencing on the MiSeq platform did not lead to quantifiable results in the proband sample, making the assay unsuitable for mosaicism detection. Therefore, individual mutation-specific PCR assays were designed and the resulting PCR products analyzed using gel electrophoresis. The individual assays' sensitivity was determined with dilution series experiments (Supplementary Note 2). Furthermore, an allele-specific PCR had to be designed for haplotyping the DNM of FAM38 in *AHDC1* due to a homopolymeric region around the mutation site for which the mutant and wildtype allele could not be phased with ONT sequencing (Supplementary Note 5 and Supplementary Data 3B).

### Long-read haplotyping assay using Oxford Nanopore Technologies (ONT)

The MinION (Oxford Nanopore Technologies [ONT], UK) long-read sequencing platform was used to determine the parent-of-origin of the DNM in the proband. To do so, primers were designed to amplify two regions (~2–16 kb each, for locations of individual primer sequences, see Supplementary Data 1) on either side of the DNM. DNA from the two parental blood samples and the diagnostic genomic DNA from the proband were amplified using LongAmp Polymerase (New England Biolabs, UK) starting with 50 ng genomic DNA in a 20 μl reaction following manufacturer's recommendations and the cycling conditions: initial 2 min at 95 °C, 30 cycles of 30 s at 95 °C and 16 min at 65 °C, and a final extension at 65 °C for 20 min. PCR amplicons were checked on a 0.9% agarose gel and if amplification had been successful, regions 1

and 2 from one sample were pooled. For library preparation, the PCR barcoding amplicon protocol and 1D ligation kit and PCR expansion kit (all ONT, UK) were used to barcode individual samples in a 20 μl PCR reaction with LongAmp polymerase, 2 μM barcoding primers and 1:100 diluted target PCR with the cycling conditions as described above for 8 cycles. After adapter ligation, the pooled library was loaded onto a MinION SpotOn Mk I version R9 flowcell (ONT) for sequencing following the manufacturer's recommendations. For initial data processing (demultiplexing and basecalling) each set of fast5 files was processed using Guppy v4.5.4 + 66c1a77 ([https://community.nanoporetech.com](https://community.nanoporetech.com)) with the parameter --config dna_r9.4.1_450bps_hac.cfg, producing one set of reads for each barcode/family member of the trio. Reads are deposited in the European Nucleotide Archive under BioProject accession number PRJEB53977 ([http://www.ebi.ac.uk/ena/data/view/PRJEB53977](http://www.ebi.ac.uk/ena/data/view/PRJEB53977)).

### Haplotype phasing of de novo mutations using Medaka and mpileup

ONT reads for each trio were aligned to the GRCh38.p12 primary assembly using minimap2 v2.18[35] with parameter -ax map-ont. Lower-quality (MAPQ < 20) and non-primary alignments were discarded using samtools view v1.12[36] with parameters -q 20 -F 256 -F 2048. For each target region (genomic coordinates are given in Supplementary Data S1), variants were called using the 'medaka_variant' workflow of Medaka v1.3.2 ([https://github.com/nanoporetech/medaka](https://github.com/nanoporetech/medaka), accessed 6th May 2021) with default parameters. The set of VCFs per region were then concatenated using BCFtools v1.12[36] to produce one VCF per BAM, subsequently annotated using dbSNP v153[37] ([https://ftp.ncbi.nih.gov/snp/latest_release/VCF/GCF_000001405.38.gz](https://ftp.ncbi.nih.gov/snp/latest_release/VCF/GCF_000001405.38.gz), accessed 6th May 2021).

Where possible, Medaka uses the information contained within heterozygous SNPs to impute the haplotype of the aligned reads. In practice this means that a proportion of the calls in each VCF are phased, being assigned to a 'phase set' of SNPs on the same haplotype. Given that the sequencing data represent mother/father/proband trios, with each proband having a DNM, each VCF was parsed to determine whether Medaka had called and phased the DNM in the proband (but not in either parent, confirming their true "de novo" status). For each DNM called by Medaka, we obtained the associated phased set SNPs, retaining only those which had a total depth of coverage >10x. We cross-referenced the phased set SNPs with the VCFs from the mother and father and identified which calls (if any) had been made at those positions. This produced a set of three haplotypes from which we used a custom script to classify the inheritance of the DNM as either maternal or paternal (the SNPs in phase with the DNM could only be derived from the chromosome inherited from the mother or father, respectively), else unresolved (Medaka either did not call the DNM in the child, called it but did not construct a phased set, or, if it did construct a phased set, either did not call its constituent SNPs in the parents or made identical calls for both of them).

DNMs not successfully phased using Medaka (Supplementary Note 4) were phased by programmatic and/or manual inspection of read pileups. A programmatic approach was implemented using a custom script which parsed read pileups (generated using samtools mpileup with parameters -aa --output-QNAME) to obtain a set of reads which contained both the ALT-allele for the DNM and a candidate phasing SNP (considered the closest one to it and for which there was a prior, namely inclusion in dbSNP). We then constructed a 2×2 count table (rows: number of reads calling REF/ALT at DNM position, columns: number of reads calling REF/ALT at phasing SNP positions) and resolved inheritance by identifying which of the two alleles for the phasing SNP, REF or ALT, were disproportionately found on the same read as the DNM ALT. Significance was assessed using Fisher's exact test. Haplotypes flagged as not programmatically resolved by either Medaka or pileup were manually reviewed using IGV v2.11.2[38], with visual inspection also used to validate all the above calls. Full details, and all scripts used for this analysis, are available at github.com/sjbush/pregcare[34]. For one family (FAM38) the phase could not be resolved with long-read sequencing due to a homopolymeric stretch around the mutation site. For this family, an allele-specific PCR was performed (Supplementary Note 5).

### Statistical analysis and data visualization
Statistical analysis was performed in R v4.2.0[39]. Figures were created using the R package ggplot2 v3.3.5[40], GraphPad Prism v9.2.0 (GraphPad Software, San Diego, California, USA, [http://www.graphpad.com](http://www.graphpad.com)) and Adobe Illustrator (Adobe Inc., [https://adobe.com/products/illustrator](https://adobe.com/products/illustrator)).

### Reporting summary
Further information on research design is available in the Nature Portfolio Reporting Summary linked to this article.

## Data availability
The sequencing data generated during the current study have been deposited in the European Nucleotide Archive under BioProject accession number PRJEB53977. Additionally, the processed sequencing data relevant to each DNM (Deep-NGS (Supplementary Data 2) and ONT calls (Supplementary Data 3A, B)) are included as Supplementary Material.

## Code availability
All code used in this study is available at github.com/sjbush/pregcare[34].

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

## Acknowledgements

This work was primarily supported by grants from NewLife (17/18/04 to A.Go.), the Wellcome (102731/Z/13/Z (to A.O.M.W.) and 219476/Z/19/Z (to A.Go.)), and the National Institute for Health Research (NIHR) Oxford Biomedical Research Centre Programme (to A.Go. and A.O.M.W.). We acknowledge WIMM core funding from the Medical Research Council (MRC) through the WIMM Strategic Alliance (G0902418 and MC_UU_12025) and the support of the NIHR UK Rare Genetic Disease Research Consortium (Musketeers' Memorandum). We thank Tim Rostron and the members of the Oxford Genomic Medicine laboratory for technical support. The funders had no role in study design, data collection and analysis, decision to publish, or preparation of the manuscript.

## Author contributions

Conceived and supervised the study, secured funding, obtained ethical approval, designed the experiments: A.O.M.W. and A.Go.; performed the experiments: M.B., U.B.A., K.A.W., and S.G.; performed data analysis: M.B., U.B.A., S.J.B., E.G., N.K., G.J.M., L.M.T., J.W., R.W.D., A.O.M.W., and A.Go.; recruited participants and/or provided samples: J.W., E.M.B., A.B., E.B.-W., N.C., A.T.De., A.Di., J.E., F.E., A.Ga., E.H., M.H., T.H., J.A.H., D.J., W.J., F.B.K., U.K., E.K., A.K., M.M.L., H.L., J.E.V.M., A.H.N., S.R., K.S., D.J.S., L.S., M.Sp., A.S., H.S., M.Su., and P.C.; designed the figures: M.B., K.A.W., S.J.B., A.O.M.W., and A.Go.; wrote the manuscript: A.O.M.W and A.Go. with input of M.B., S.J.B., L.M.T., and R.W.D.

## Competing interests

The authors declare no competing interests.

## Additional information

[1]Clinical Genetics Group, MRC Weatherall Institute of Molecular Medicine, Radcliffe Department of Medicine, University of Oxford, Oxford, UK. [2]NIHR Oxford Biomedical Research Centre, Oxford, UK. [3]St. Anna Children's Cancer Research Institute (CCRI), Vienna, Austria. [4]Victor Chang Cardiac Research Institute, Darlinghurst, NSW, Australia. [5]Centre for Population Genomics, Garvan Institute of Medical Research, UNSW Sydney, Sydney, NSW, Australia. [6]Oxford Genetics Laboratories, Churchill Hospital, Oxford University Hospitals NHS Foundation Trust, Oxford, UK. [7]Oxford Centre for Genomic Medicine, Nuffield Orthopaedic Centre, Oxford University Hospitals NHS Foundation Trust, Oxford, UK. [8]Manchester Centre for Genomic Medicine, Manchester University NHS Foundation Trust, Manchester, UK. [9]Division of Evolution and Genomic Sciences, University of Manchester, Manchester, UK. [10]Department of Clinical Genetics, Liverpool Women's NHS Foundation Trust, Liverpool, UK. [11]Clinical Genetics Department, Guy's Hospital, Guy's & St Thomas' NHS Foundation Trust, London, UK. [12]Nottingham Regional Genetics Service, City Hospital Campus, Nottingham University Hospitals NHS Trust, Nottingham, UK. [13]South West Thames Regional Genetics Service, St George's University Hospitals NHS Foundation Trust, London, UK. [14]North West Thames Regional Genetics Service, London North West University Healthcare NHS Trust, Northwick Park Hospital, Harrow, UK. [15]North East Thames Regional Genetics Service, Great Ormond Street Hospital NHS Foundation Trust, London, UK. [16]Sheffield Clinical Genetics Service, Sheffield Children's NHS Foundation Trust, Sheffield, UK. [17]Clinical Genetics, Royal Devon & Exeter Hospital (Heavitree), Royal Devon University Healthcare NHS Foundation Trust, Exeter, UK. [18]MRC London Institute of Medical Sciences, Institute of Clinical Sciences, Faculty of Medicine, Imperial College London, London, UK. [19]West Midlands Regional Clinical Genetics Service and Birmingham Health Partners, Birmingham Women's and Children's Hospitals NHS Foundation Trust, Birmingham, UK. [20]Nuffield Department of Clinical Neurosciences, University of Oxford, Oxford, UK. [21]Wessex Clinical Genetics Service, University Hospital Southampton, Princess Anne Hospital, Southampton, UK. [22]Northern Genetics Service, The Newcastle upon Tyne Hospitals NHS Foundation Trust, Newcastle, UK. [23]Department of Statistics, University of Oxford, Oxford, UK. [24]These authors contributed equally: Marie Bernkopf, Ummi B. Abdullah. [25]These authors jointly supervised this work: Andrew O.M. Wilkie, Anne Goriely. ✉e-mail: anne.goriely@imm.ox.ac.uk

