## [Peer Review File · Nature Communications]

Personalized recurrence risk assessment following the birth of a child with a pathogenic de novo mutationEditorial Note: This manuscript has been previously reviewed at another journal that is not operating a transparent peer review scheme. This document only contains reviewer comments and rebuttal letters for versions considered at *Nature Communications* .

REVIEWERS' COMMENTS

Reviewer #1 (Remarks to the Author):

Thanks to the Authors for addressing my comments.

Please indicate in the abstract how many genes were targeted for ultra deep short-read sequencing.

Reviewer #2 (Remarks to the Author):

I have no more comments. Only a suggestion to add a statement in Discussion that the current long read technologies allow determination of parental origin of a variant by studying the proband sample only.

Akbari et al. Parent-of-origin detection and chromosome-scale haplotyping using long-read DNA methylation sequencing and Strand-seq. doi: <https://doi.org/10.1101/2022.05.24.493320>
<https://www.biorxiv.org/>

REVIEWERS' COMMENTS

Reviewer #1 (Remarks to the Author):

Thanks to the Authors for addressing my comments.

Please indicate in the abstract how many genes were targeted for ultra deep short-read sequencing.

We thank the reviewer for taking the time to re-review our manuscript and consider our responses .

We note that the reviewer's suggestion – i.e. to add the number of genes targeted in this study - was in fact included in the version of the abstract we had submitted and which read:

“We recruited 60 families with one (n = 58) or more (n = 2) offspring affected by a serious developmental disorder caused by an apparent DNMs (total of 61 DNMs in 51 different genes) and used targeted ultra-deep sequencing of multiple tissues from the mother-father-child trio to identify cases of occult mosaicism.”

However, given that we have been asked by the Editor to shorten to the abstract to ~150 words, we do not have the space to keep this sentence in full in the latest version and we now only mentioned the number of DNMs and genes that were used for quantification.

This part of the abstract now reads:

“Among 58 families with a single affected offspring (representing 59 *de novo* mutations in 49 genes), the recurrence risk for 35 (59%) was decreased below 0.1%, but increased owing to parental mixed mosaicism for 5 (9%) – that could be quantified in semen for paternal cases (recurrence risks of 5.6-12.1%).”

Reviewer #2 (Remarks to the Author):

I have no more comments. Only a suggestion to add a statement in Discussion that the current long read technologies allow determination of parental origin of a variant by studying the proband sample only.

Akbari et al. Parent-of-origin detection and chromosome-scale haplotyping using long-read DNA methylation sequencing and Strand-seq. doi: <https://doi.org/10.1101/2022.05.24.493320>
<https://www.biorxiv.org/>

We thank the reviewer for taking the time to re-read the manuscript and for the helpful suggestion of this recent reference, which is relevant to the manuscript and has been added to the discussion (ref 27) and to the supplementary notes (ref 22).